# Post-Conization HPV Vaccination and Its Impact on Viral Status: A Retrospective Cohort Study in Troms and Finnmark, 2022

**DOI:** 10.3390/pathogens13050381

**Published:** 2024-05-02

**Authors:** Marie Rykkelid, Helga Marie Wennberg, Elin Richardsen, Sveinung Wergeland Sørbye

**Affiliations:** 1Department of Clinical Medicine, Faculty of Health Sciences, UiT The Arctic University of Norway, 9019 Tromsø, Norway; mry044@post.uit.no (M.R.); hwe024@uit.no (H.M.W.); 2Department of Medical Biology, Faculty of Health Sciences, UiT The Arctic University of Norway, 9019 Tromsø, Norway; elin.richardsen@unn.no; 3Department of Clinical Pathology, University Hospital of North Norway (UNN), 9019 Tromsø, Norway

**Keywords:** human papillomavirus (HPV), HPV vaccine, conization, LEEP, recurrence, post-conization recurrence, HPV status, cervical cancer prevention, vaccination efficacy, cervical intraepithelial neoplasia

## Abstract

Human papillomavirus (HPV) is associated with cellular changes in the cervix leading to cancer, which highlights the importance of vaccination in preventing HPV infections and subsequent cellular changes. Women undergoing the loop electrosurgical excision procedure (LEEP), a treatment for high-grade cervical intraepithelial neoplasia (CIN2+), remain at risk of recurrence. This study assessed the effect of post-conization HPV vaccination on the viral status of women at six months post-conization, aiming to evaluate the vaccine’s effectiveness in preventing recurrence of CIN2+. A retrospective cohort study was conducted among women in Troms and Finnmark who underwent conization in 2022. Using the SymPathy database and the national vaccination register (SYSVAK), we analyzed the vaccination statuses and HPV test results of women born before 1991, who had not received the HPV vaccine prior to conization. Out of 419 women undergoing conization, 243 met the inclusion criteria. A significant association was found between post-conization HPV vaccination and a negative HPV test at six months of follow-up (ARR = 12.1%, *p* = 0.039). Post-conization HPV vaccination significantly reduced the risk of a positive HPV test at the first follow-up, suggesting its potential in preventing the recurrence of high-grade cellular changes. However, the retrospective design and the insufficient control of confounding variables in this study underscore the need for further studies to confirm these findings.

## 1. Introduction

In Norway, the National Cervical Screening Program aims to prevent cervical cancer through early identification and treatment of high-grade cervical intraepithelial neoplasia (CIN2+) before it progresses to malignancy and to reduce the incidence and mortality of cervical cancer [1,2]. The program, managed by the National Cancer Registry of Norway, was established as a national initiative in 1995 and targets women aged 25 to 69.

As of July 2023, women now undergo high-risk HPV testing every five years, where positive results are followed up with microscopic examination of the cervical smear [3,4]. This represents a shift from the prior triennial microscopic evaluations that contributed to the observed decline in cervical cancer mortality since the 1980s [2]. Notably, in Norway, more than half of cervical cancer cases arise in individuals who were non-compliant with the screening recommendations.

Persistent infection with HPV can cause cervical intraepithelial neoplasia (CIN) [5], also known as cervical dysplasia or precancerous abnormal cells, and more than 99% of cervical cancers are linked to HPV [6]. The conventional management of CIN2+ involves conization, specifically the loop electrosurgical excision procedure (LEEP), which removes a lesion that has the potential to progress to cervical cancer [7]. A follow-up examination post-conization is scheduled approximately six months later and includes an HPV test to inform further treatment [8]. In 2022, 6393 women underwent conization in Norway with an average age of 39.6 years, and the median age was 36 years [9]. Conization poses risks, such as an elevated likelihood of preterm births [10,11], highlighting the relevance of fertility considerations in this patient group. Preventing recurrence of CIN2+ in treated women will result in the prevention of potential re-conizations or hysterectomies, reducing the impact on fertility.

The HPV vaccine was introduced to the Norwegian childhood vaccination program in 2009 for seventh-grade girls (born from 1997 onwards), and was extended to boys of the same grade in 2018 (born from 2006 onwards) [12]. A catch-up program from 2016 to 2018 offered the vaccine to unvaccinated women born after 1990, which achieved vaccination with at least one dose in 59% of women born between 1991 and 1996 [13]. By 2022, vaccination coverage reached 93% among 16-year-old girls. This extensive vaccination effort has not only increased coverage among the youth but also indirectly increased herd immunity against HPV.

There are two different HPV vaccines available in Norway: Cervarix, which is utilized in the national vaccination program and covers HPV types 16 and 18, and Gardasil 9, which in addition to HPV types 16 and 18 also covers types 31, 33, 45, 52, and 58 and genital warts caused by types 6 and 11 [14]. Cervarix offers cross-protection against additional HPV types beyond 16 and 18. Despite their inability to treat existing infections, these vaccines effectively prevent new HPV infections. Cervarix and Gardasil 9 demonstrate over 90% and up to 100% efficacy, respectively, against the development of CIN2+ associated with the vaccine’s covered HPV types [12].

Emerging research suggests that the HPV vaccine has the potential to prevent recurrence of high-grade cervical dysplasia post-conization [15,16,17,18], and yet there is a lack of consensus or national guidelines endorsing post-treatment HPV vaccination for CIN2+ [19]. Consequently, the decision to recommend vaccination post-conization is left to individual gynecologists, which, combined with out-of-pocket costs [14], leads to disparities in vaccine uptake among treated women.

This study aimed to evaluate the impact of post-conization HPV vaccination on the HPV status at the first follow-up six months post-treatment. Considering the link between HPV infections and over 99% of cervical dysplasia and cancers [6,20], this study sought to indirectly determine the vaccine’s role in mitigating the recurrence risk of CIN2+ by running a comparison of HPV status between vaccinated and unvaccinated women post-conization.

## 2. Materials and Methods

### 2.1. Study Design

This retrospective cohort study was performed at the University Hospital of North Norway (UNN), Tromsø. Inclusion criteria were women who underwent conization in Troms and Finnmark during the year 2022. Exclusion criteria included the absence of a post-conization HPV test, HPV vaccination prior to conization, and birth after the year 1990.

The primary outcome measure, to assess the HPV vaccine’s impact, was the HPV status at the initial follow-up six months after conization. The exclusion of women vaccinated prior to conization ensured the separation of the vaccine’s effects pre- and post-procedure. Additionally, the exclusion of women born after 1990 was due to their eligibility for free vaccination under the national childhood vaccination or catch-up programs, leading to higher vaccination rates in this cohort compared to those born before 1991.

### 2.2. Data Collection Methodology

In this study, we utilized the SymPathy database, a comprehensive Laboratory Information System (LIS) employed by the Department of Clinical Pathology at the University Hospital of North Norway (UNN). SymPathy serves as the central repository for all cervical cytology, histology, and HPV test specimens from women in Troms and Finnmark counties, representing approximately 5% of Norway’s total population. This database is essential for the systematic registration, analysis, and management of patient data, playing a crucial role in the region’s cervical cancer prevention efforts. For our research, SymPathy was instrumental in identifying women born before 1991 who underwent conization during 2022. These patients’ cervical tissue samples were meticulously recorded and evaluated by department pathologists, adhering to standard operational procedures. Conization, primarily executed via the loop electrosurgical excision procedure (LEEP), resulted in tissue specimens being cataloged with the NORPAT code T83701, denoting cervix cone topography.

The SymPathy database aggregated extensive data on each patient, including identification numbers, historical screening outcomes, therapeutic actions, and follow-up evaluations featuring cytology and HPV test results. While Norwegian guidelines recommend co-testing with both cytology and an HPV test at the first follow-up after treatment, this study focused solely on the post-conization HPV results as the study endpoint. By leveraging the NORPAT code, we could efficiently extract data relevant to individuals treated with conization, encompassing HPV test outcomes at 6 months post-procedure. The UNN uses the Roche Cobas 4800 system for HPV testing, which performs individual assessments for HPV types 16 and 18 and a pooled analysis for twelve additional HPV types (HPV 31, 33, 35, 39, 45, 51, 52, 56, 58, 59, 66, and 68). This meticulous approach to data collection via SymPathy underscores the depth and reliability of our study’s dataset.

### 2.3. HPV Vaccination Status Assessment

To ascertain the HPV vaccination status post-conization, data were retrieved from the national vaccination registry (SYSVAK) through a meticulous manual search process using the patient identification numbers. Vaccination status was classified into three distinct groups: (1) vaccinated prior to conization, (2) vaccinated subsequent to conization, and (3) unvaccinated individuals. Patients who received an HPV vaccine both before and after conization were accounted for in both vaccinated groups. No differentiation was made based on the vaccine variant; any recipient of at least one vaccine dose was considered vaccinated. The timing of post-conization vaccination was not explicitly analyzed relative to HPV testing, but it is presumed that the majority were vaccinated before HPV testing at the six-month follow-up.

### 2.4. Ensuring Data Privacy Integrity

In adherence with privacy regulations and ensuring patient confidentiality, data collection was exclusively conducted by the principal investigator (SWS), a senior consultant in pathology at the Department of Clinical Pathology, UNN. The anonymization process entailed the elimination of identifiable personal information, including names and identification numbers, to comply with ethical standards. Consequently, only anonymized frequency tables were provided to the authors for detailed analysis and manuscript development.

### 2.5. Statistical Evaluation

The data analysis was conducted using SPSS software, version 29.0.1.0., with the study cohort consisting of females born before 1991 who were unvaccinated prior to conization and had available HPV test results from the follow-up approximately six months after the procedure. Based on the defined inclusion criteria, frequency distribution tables were generated to illustrate the age-specific HPV status and vaccination status, alongside the correlation between post-conization HPV vaccination and subsequent HPV test outcomes. Statistical significance was established at *p* < 0.05, with the number needed to vaccinate (NNV) in order to prevent a single positive HPV test and the absolute risk reduction (ARR) being calculated manually.

## 3. Results

In 2022, conization specimens from 419 women residing in Troms and Finnmark were processed by the Department of Clinical Pathology at the University Hospital of North Norway (UNN). A subset of these women, specifically 25, did not undergo an HPV test post-conization for reasons including subsequent hysterectomy, treatment for cervical cancer with radiochemotherapy, or absence of follow-up at the time of data collection. From the remaining 394 women who did have an HPV test, 141 born after 1990 were excluded from this study due to their eligibility for prior vaccination programs. An additional 10 women were excluded for having received the HPV vaccine before their conization, despite not being eligible for the initial childhood vaccination or catch-up programs. Consequently, a total of 243 women met the criteria for inclusion in this study (Figure 1).

The distribution of HPV status within the study population, segmented by age, is detailed in Table 1. Participants were categorized into age groups spanning 10 years, with the youngest group being 30–39 years and the oldest 70–79 years. Within each age group, women were divided into two categories based on their post-conization HPV test results: HPV+ for positive tests and HPV− for negative tests.

The distribution of women across age groups showed a concentration in the youngest category (*n* = 108), with a progressive decrease in the number of participants in each subsequent older age group. Specifically, the age groups of 60–69 and 70–79 years comprised 16 and 8 participants, respectively.

Approximately six months following conization, the average rate of HPV positivity among participants younger than 60 years was 26.9%. In contrast, in the oldest age cohorts, a higher prevalence of 75% HPV positivity was observed after conization. The disparity in HPV positivity rates between participants younger than 60 years and those aged 60 and above was statistically significant (*p* < 0.001, Fisher’s exact test, one-sided).

The analysis of vaccination coverage revealed that the highest rates were among the three youngest age groups. Specifically, the age groups 30–39 and 50–59 years demonstrated similar vaccination rates of 37.0% and 36.7%, respectively, whereas the 40–49 age group exhibited a vaccination rate of 27.4%. Notably, vaccination coverage was markedly lower in the oldest age groups, with 12.5% in the 60–69 age group and 0% in the 70–79 age group.

Table 2 outlines the categorization of participants based on whether they received the HPV vaccine post-conization (YES or NO) and their HPV test results approximately six months after the procedure (HPV+ or HPV−).

Out of the 243 participants included in this study, 77 had been vaccinated post-conization. This represents less than one-third of the sample. A considerable proportion, 77 out of 243 (31.7%), tested positive for HPV at about six months post-conization. The incidence of HPV positivity was 23.4% in the vaccinated group compared to 35.5% in the unvaccinated group, resulting in an absolute risk reduction (ARR) of 12.1 percentage points. The significance of the vaccination’s effect was evaluated using a one-sided Fisher’s exact test, yielding a *p*-value of 0.039, which indicates a significant correlation between post-conization vaccination and a negative HPV test at the six-month follow-up. The number needed to vaccinate (NNV) to prevent a single positive HPV test six months post-conization was calculated to be 8.22.

## 4. Discussion

Evidence from multiple studies indicates that post-conization HPV vaccination may play a preventative role in the recurrence of high-grade intraepithelial neoplasia (CIN2+) [16,17,18]. Our research identified a notable reduction in HPV positivity rates at six months after conization, from 35.5% in the unvaccinated group to 23.4% in the vaccinated group. This translates to an absolute risk reduction (ARR) of 12.1 percentage points in the likelihood of testing positive for HPV among vaccinated individuals compared to their unvaccinated counterparts. This outcome not only underlines the vaccine’s potential in reducing the likelihood of future CIN2+ but also its role in mitigating cancer risk due to HPV.

The disparity in conization rates across age demographics can likely be linked to the prevalence of CIN2+ within the population. Data from Norway in 2022 reveal a median conization age of 36 years, with a noteworthy observation that only 149 out of 6393 women who underwent conization were aged 70 or above [9]. This trend suggests a decrease in conization procedures with advancing age. Notably, our study found the highest HPV positivity rates post-conization among the eldest cohorts. A contributing factor might be the minimal vaccination coverage observed in these groups, with only 2 out of 24 (8.3%) women aged 60 or above receiving the HPV vaccine post-conization. An alternative explanation could be the diminished immune response and weaker immune system prevalent in older populations [21]. However, the small sample size warrants caution in drawing definitive conclusions regarding this observed difference.

Less than one-third of the study population opted for vaccination post-conization. This phenomenon is potentially due to the lack of explicit national guidelines on post-conization HPV vaccination. While our analysis did not delve into how economic, educational, or other socio-economic factors might influence vaccination decisions, the high cost of the vaccine suggests such variables could impact vaccination rates. The significant correlation between HPV vaccination and a negative HPV test post-conization confirms the vaccine’s efficacy in lowering positive HPV test occurrences following conization. Nonetheless, this study’s brief follow-up time limits our ability to link HPV test outcomes to the incidence of diagnosed recurrences of CIN2+. Consequently, the HPV status serves merely as an indirect indicator of the vaccine’s effectiveness against CIN2+ recurrence. A comprehensive evaluation of the vaccine’s long-term impact necessitates extended follow-up periods.

Chen et al.’s prospective cohort study, with a 2-year follow-up, investigated the vaccine’s influence on CIN2+ recurrence [15]. The analysis, which included HPV testing at 6 months, 1 year, and 2 years, defined recurrence through histologically confirmed CIN2+. They found a recurrence rate of 10.6% in the unvaccinated group compared to 2.0% in the vaccinated group, demonstrating a significant reduction in CIN2+ recurrence among those who received the vaccine (*p* = 0.001). The severity of cervical dysplasia at conization and the HPV type detected prior to conization were considered confounding factors. Although initial 6-month findings showed no significant difference in HPV positivity rates between the vaccinated and unvaccinated groups, a substantial decrease in positive tests was noted in the vaccinated group at the 2-year mark (8.1% versus 15.8%, *p* = 0.026). This evidence suggests that the vaccine’s efficacy against HPV may increase over time, indicating the potential of observing a greater preventative effect in our study with a prolonged observation period.

Various studies, inclusive of meta-analyses and a randomized controlled trial (RCT), have substantiated the preventive efficacy of the HPV vaccine against the recurrence of CIN2+ [18,22,23]. Notably, one such meta-analysis [23] demonstrated a significant risk reduction for the development of new high-grade intraepithelial lesions after HPV vaccination, with a relative risk of 0.41 (95% CI [0.27; 0.64]), underscoring the broad protective effects of vaccination across different patient demographics and HPV types. This aligns with our findings, which also suggest a reduced risk of recurrence as indicated by a higher proportion of negative HPV tests in the vaccinated cohort.

Furthermore, the meta-analysis by Jentschke et al. included diverse study designs—three retrospective, three prospective studies, three post hoc analyses of RCTs, and one cancer registry study—highlighting the consistency of vaccine efficacy across various research settings. This robust collection of data underscores the vaccine’s potential protective effect post-conization, with the studies included frequently utilizing histologically verified CIN2+ as an evaluative endpoint.

Our investigation, while focusing on HPV test results to assess the post-conization vaccine impact, found similar protective trends. According to Kreimer et al., the HPV test used in our study exhibits a 99% negative predictive value for CIN2+ [24], suggesting that the increased proportion of women with a negative HPV test within the vaccinated cohort is a suitable indicator for diminished risk of post-conization recurrence of CIN2+. This corroborates the observed prophylactic vaccine benefit as reported in the broader literature and specifically in the detailed analysis provided as part of the meta-analysis by Jentschke et al.

In this research, we explored the impact of post-conization HPV vaccination on HPV test results at six months following the treatment. While acknowledging that numerous studies have addressed similar topics—some with extended follow-up periods and more comprehensive data collection—it is important to contextualize our findings within the existing body of literature. Several studies have indeed demonstrated the efficacy of HPV vaccination in mitigating the recurrence of high-grade lesions (CIN2+) post-treatment; however, the vaccination of adult women who are already HPV-positive remains a contentious issue. The majority of these studies are non-randomized and involve relatively small cohorts. Our study contributes additional evidence supporting the administration of prophylactic HPV vaccines to reduce the risks of persistent HPV infections and recurrence post-conization, even though it also shares the non-randomized nature and scale limitations typical of this research area.

Our analysis gains particular relevance from its setting within the Norwegian healthcare framework. The high coverage of HPV vaccination among younger women, especially those born post-1997 due to an established national vaccination program starting in 2009, contrasts with the demographic of our study—women aged 30 years and older. This age group benefits less from historical vaccination efforts and represents a population for whom the benefits of post-conization HPV vaccination are less documented and more debated. This focus on an older demographic, typically underrepresented in vaccination studies, brings a unique perspective to the discourse on the effectiveness of HPV vaccination in preventing recurrence, thereby enriching the overall understanding and strategic discussions surrounding HPV management in diverse healthcare settings.

Nonetheless, our study has limitations, including a modest sample size and a skewed ratio of vaccinated to unvaccinated participants. Moreover, this study witnessed an uneven age distribution, coupled with a significant variance in HPV test results across the youngest and eldest age brackets, which established age as a confounding variable. Additionally, this study’s retrospective format diminished our capacity to adjust for other potential confounders due to the non-collection of data such as socio-economic status, sexual partnership history, smoking habits, or immunosuppressive conditions. Moreover, the correlation between the severity of the cervical dysplasia and positive post-conization HPV tests was not explored, nor whether the entirety of the area with cervical dysplasia was excised during conization (ensuring clear resection margins).

Furthermore, this investigation did not meticulously analyze the timing of vaccination in relation to conization, potentially leading to imprecise classifications regarding vaccination status. While the assumption was that the majority were vaccinated before the initial HPV test at six months post-conization, it is possible that a subset was vaccinated afterward. This could potentially have clouded the accuracy of our assessment of the vaccine’s true efficacy. Nonetheless, given the observed effect of the vaccine, it is reasonable to infer that most participants were likely vaccinated before their first post-conization HPV test. A more detailed examination of the vaccination timeline might have provided stronger support for this study’s conclusions. Lastly, this study did not consider the specific HPV vaccine types used or the number of doses administered prior to the HPV test, limiting our ability to evaluate whether vaccines against a broader spectrum of HPV types or a higher number of doses might lead to more negative HPV tests.

## 5. Conclusions

This study’s findings indicate that administering the HPV vaccine after conization significantly increases the likelihood of a negative HPV test at the six-month follow-up. This emphasizes the vaccine’s potential utility in diminishing the risk of recurrence for high-grade intraepithelial neoplasia (CIN2+) post-treatment, corroborating evidence presented in previous research. However, the limited sample size, short follow-up duration, and this study’s retrospective nature, which restricted a comprehensive evaluation of all potential confounding variables, highlight the need for further empirical inquiry to substantiate these preliminary observations. The call for additional research is particularly pressing in light of the absence of national guidelines for post-conization HPV vaccination, despite a growing evidence base that supports its implementation. The current scarcity of randomized controlled trials examining the vaccine’s effect in this context underscores the critical need for ongoing and future studies to definitively establish the vaccine’s impact on post-conization HPV status and recurrence of CIN2+.

## Figures and Tables

**Figure 1 pathogens-13-00381-f001:**
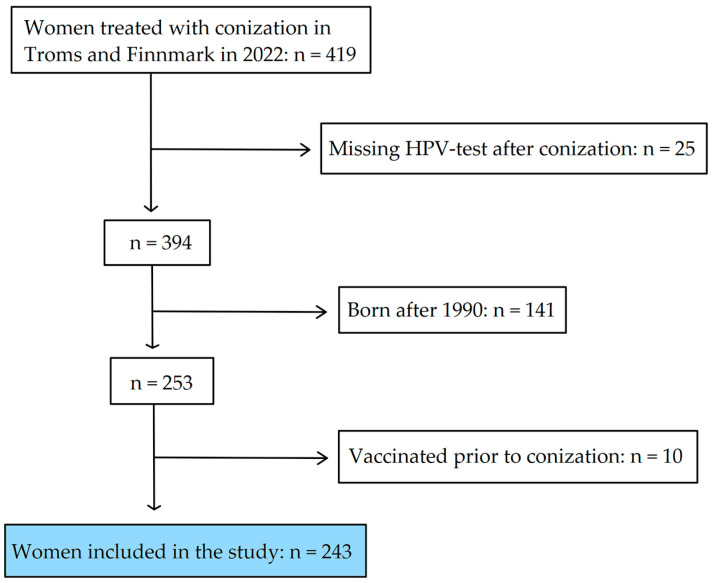
Flow-chart of the inclusion process of treated women in this study.

**Table 1 pathogens-13-00381-t001:** Women treated with conization included in this study segmented by age. Each age bracket is divided into categories based on HPV test results at the six-month follow-up and how many in each bracket received at least one dose of an HPV vaccine. Percentages of HPV-positive women and vaccinated women are also given.

Age Group	HPV− (*n* = 166)	HPV+ (%) (*n* = 77, 31.7%)	Total (*n* = 243)	Vaccinated (%) (*n* = 77, 31.7%)
30–39 years	78	30 (27.8)	108	40 (37.0)
40–49 years	47	15 (24.2)	62	17 (27.4)
50–59 years	35	14 (28.6)	49	18 (36.7)
60–69 years	4	12 (75.0)	16	2 (12.5)
70–79 years	2	6 (75.0)	8	0 (0.0)

**Table 2 pathogens-13-00381-t002:** Frequency table that compares the HPV vaccination status (no/yes) with the result of the HPV test taken at the six-month follow-up (HPV−/HPV+).

Vaccinated (after Conization)	HPV− (*n* = 166)	HPV+ (%) (*n* = 77, 31.7%)	Total (*n* = 243)	*p*-Value
No	107	59 (35.5)	166	0.039
Yes	59	18 (23.4)	77	

## Data Availability

The raw data supporting the conclusions of this article will be made available by the authors on request.

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
