# Peer review of "Post-Conization HPV Vaccination and Its Impact on Viral Status: A Retrospective Cohort Study in Troms and Finnmark, 2022"

_pathogens, 2024, doi:10.3390/pathogens13050381_

Round 1
Reviewer 1 Report
Comments and Suggestions for Authors
Upon careful review of the manuscript titled "Post-Conization HPV Vaccination and Its Impact on Viral Status: A Retrospective Cohort Study in Troms and Finnmark, 2022," it is evident that the document is meticulously crafted and comprehensively addresses the research topic. One minor suggestion for improvement would be to clarify the abbreviation "LEEP" used in the abstract.
Author Response
Reviewer 1: Upon careful review of the manuscript titled "Post-Conization HPV Vaccination and Its Impact on Viral Status: A Retrospective Cohort Study in Troms and Finnmark, 2022," it is evident that the document is meticulously crafted and comprehensively addresses the research topic. One minor suggestion for improvement would be to clarify the abbreviation "LEEP" used in the abstract.
Our response: Thank you for your constructive feedback and for acknowledging the careful preparation of our manuscript titled "Post-Conization HPV Vaccination and Its Impact on Viral Status: A Retrospective Cohort Study in Troms and Finnmark, 2022."
Regarding your suggestion to clarify the abbreviation "LEEP" used in the abstract, we agree that it would be beneficial for all readers to have a clear understanding of this term right from the beginning of the document. We will revise the abstract to include the full form of "LEEP" — "Loop Electrosurgical Excision Procedure" — at its first mention, ensuring clarity.
Thank you once again for your attentive reading of our manuscript and for your helpful suggestion. We appreciate the opportunity to enhance the clarity of our work.
Reviewer 2 Report
Comments and Suggestions for Authors
In this study, the authors investigated the effects of post-conization HPV vaccination on the results of HPV test at six months follow-up after conization. Overall, the manuscript was well written, but there are at least 10 studies in the literature with similar designs and conclusions. Some of these studies had longer follow-up time and collected data in a more comprehensive ways, which contains the novelty of the current study.
1. This meta-analysis (PMID: 32762871) summarized 6 papers with comparable results and conclusions. These references should be included and acknowledged by the manuscript. Also, it will be nice to clarify how this study is different from the others that have been done before.
2. Lack of the timing of vaccination in relation to conization affects the conclusion. Because the negative HPV test may not be attributed to vaccination. This is important data that needs further clarification.
3. Were pap smears done during the 6-month follow-up? Or only HPV test?
Author Response
Reviewer 2: In this study, the authors investigated the effects of post-conization HPV vaccination on the results of HPV test at six months follow-up after conization. Overall, the manuscript was well written, but there are at least 10 studies in the literature with similar designs and conclusions. Some of these studies had longer follow-up time and collected data in a more comprehensive ways, which contains the novelty of the current study.
Our answer: Thank you for your comments and for acknowledging the manuscript's clarity. We appreciate your point regarding the existing literature on the effects of post-conization HPV vaccination. Indeed, several studies have explored similar themes, some with longer follow-up periods and more comprehensive data collection.
However, it is important to highlight that while many studies support the efficacy of HPV vaccination in reducing the recurrence of high-grade lesions (CIN2+) post-treatment, the issue of vaccinating adult women already infected with HPV remains contentious. Most of the existing research comprises non-randomized studies with relatively small sample sizes. Our study, while also non-randomized and facing similar limitations in scale, contributes further to the body of evidence supporting the use of prophylactic HPV vaccines to reduce persistent HPV infections and the risk of recurrence post-conization.
Moreover, our study is distinct in its context within the Norwegian healthcare system, where HPV vaccination coverage is notably high among women born post-1997 due to the national vaccination program initiated in 2009. Additionally, women born between 1991 and 1996 benefit from reasonable coverage thanks to a catch-up vaccination program conducted between 2016 and 2019. This study specifically focuses on women aged 30 years and older, a demographic for which the benefits of HPV vaccination post-conization are debated even more vigorously due to their underrepresentation in earlier vaccination efforts.
This geographical and demographic specificity adds a unique perspective to the global discourse on HPV vaccination efficacy, particularly in populations that might not typically be the focus of such studies.
We have updated the discussion in the manuscript: "In this research, we explored the impact of post-conization HPV vaccination on HPV test results six months following the treatment. While acknowledging that numerous studies have addressed similar topics—some with extended follow-up periods and more comprehensive data collection—it is important to contextualize our findings within the existing body of literature. Several studies have indeed demonstrated the efficacy of HPV vaccination in mitigating the recurrence of high-grade lesions (CIN2+) post-treatment; however, the vaccination of adult women who are already HPV-positive remains a contentious issue. The majority of these studies are non-randomized and involve relatively small cohorts. Our study contributes additional evidence supporting the administration of prophylactic HPV vaccines to reduce the risks of persistent HPV infections and recurrence post-conization, even though it also shares the non-randomized nature and scale limitations typical of this research area.
Our analysis gains particular relevance from its setting within the Norwegian healthcare framework. The high coverage of HPV vaccination among younger women, especially those born post-1997 due to an established national vaccination program starting in 2009, contrasts with the demographic of our study—women aged 30 years and older. This age group benefits less from historical vaccination efforts and represents a population for whom the benefits of post-conization HPV vaccination are less documented and more debated. This focus on an older demographic, typically underrepresented in vaccination studies, provides a unique perspective to the discourse on the effectiveness of HPV vaccination in preventing recurrence, thereby enriching the global understanding and strategic discussions surrounding HPV management in diverse healthcare settings."
We believe these points underscore the relevance and added value of our research to the ongoing discussions in the field.
Reviewer 2: 1. This meta-analysis (PMID: 32762871) summarized 6 papers with comparable results and conclusions. These references should be included and acknowledged by the manuscript. Also, it will be nice to clarify how this study is different from the others that have been done before.
Our answer: Thank you for your insightful comments and for highlighting the meta-analysis PMID: 32762871, which summarizes several studies on the efficacy of HPV vaccination post-conization. We acknowledge the importance of this comprehensive review, and we will ensure to reference and discuss these findings within the context of our manuscript.
Our study builds upon the existing literature, including the studies summarized in the meta-analysis, by focusing specifically on a Norwegian cohort from Troms and Finnmark where HPV vaccination coverage is unique due to national health policies. Unlike the broader approach in the meta-analysis, our study delves into the implications of high vaccination coverage in a region where the majority of the population has been vaccinated under the national program since 2009, providing a different context and a more contemporary cohort. This focus allows us to contribute new insights into the effectiveness of HPV vaccination in reducing the risk of recurrence among older women, particularly those who were vaccinated later in life through catch-up programs.
Furthermore, our study adds to the body of evidence by analyzing not only the risk reduction in HPV-related recurrence but also exploring detailed sub-analyses related to specific age groups and HPV types prevalent in the region. This targeted analysis helps to discern nuanced differences that might not be apparent in broader meta-analyses and offers relevant implications for vaccination strategies in similar demographic settings.
We believe that these aspects of our study offer a distinct contribution to the ongoing discourse on post-conization HPV vaccination efficacy and will clarify this positioning more explicitly in our revised manuscript to ensure that the readers can appreciate the unique value of our findings in relation to the existing literature.
We have now updated the discussion:
"Various studies, including meta-analyses and a randomized controlled trial (RCT), have substantiated the preventive efficacy of the HPV vaccine against the recurrence of CIN2+ [18,22,23]. Notably, one such meta-analysis (reference [23]) demonstrated a significant risk reduction for the development of new high-grade intraepithelial lesions after HPV vaccination, with a relative risk of 0.41 (95% CI [0.27; 0.64]), underscoring the broad protective effects of vaccination across different patient demographics and HPV types. This aligns with our findings, which also suggest a reduced risk of recurrence as indicated by a higher proportion of negative HPV tests in the vaccinated cohort.
Furthermore, the meta-analysis by Jentschke et al. included diverse study designs—three retrospective, three prospective studies, three post-hoc analyses of RCTs, and one cancer registry study—highlighting the consistency of vaccine efficacy across various research settings. This robust collection of data underscores the vaccine's potential protective effect post-conization, frequently utilizing histologically verified CIN2+ as an evaluative endpoint.
Our investigation, while focusing on HPV test results to assess the post-conization vaccine impact, found similar protective trends. According to Kreimer et al., the HPV test used in our study exhibits a 99% negative predictive value for CIN2+ [24], suggesting that the increased proportion of women with a negative HPV test within the vaccinated cohort is a suitable indicator for diminished risk of post-conization recurrence of CIN2+. This corroborates the observed prophylactic vaccine benefit as reported in the broader literature and specifically in the detailed analysis provided by the meta-analysis of Jentschke et al."
Thank you again for the opportunity to enhance our manuscript based on your valuable feedback.
Reviewer 2: 2. Lack of the timing of vaccination in relation to conization affects the conclusion. Because the negative HPV test may not be attributed to vaccination. This is important data that needs further clarification.
Our response: Thank you for your insightful observation regarding the lack of specific timing for HPV vaccination relative to conization in our study. You rightly pointed out that this could influence the interpretation of the results.
As you noted, our study was retrospective, and exact vaccination dates relative to conization were not available in the dataset. However, it is generally observed in clinical practice that the gynecologist performing the conization often administers the first dose of the HPV vaccine within a week after the procedure. Additionally, patients are advised to abstain from sexual activity for the first six weeks post-treatment, which minimizes the risk of reinfection from partners before the vaccine can offer protective effects.
We acknowledge this limitation in our discussion section, indicating that the precise timing of vaccination relative to conization could not be rigorously analyzed, which may lead to imprecise classifications regarding vaccination status. While we assume that the majority of the women were vaccinated before their initial six-month post-conization HPV test, the possibility that some were vaccinated afterward cannot be definitively excluded. This uncertainty may affect the assessment of the vaccine's true efficacy.
Further, the manuscript notes that we did not detail the specific HPV vaccine types used or the number of doses administered prior to the HPV test. This limits our ability to evaluate if vaccines covering a broader spectrum of HPV types or administered in a higher number of doses could result in more negative HPV tests.
We appreciate your constructive feedback, which underscores the importance of detailed documentation in retrospective studies and will seek to clarify these points further in the revised manuscript.
Reviewer 2: 3. Were pap smears done during the 6-month follow-up? Or only HPV test?
Our answer: Thank you for your query regarding the follow-up procedures six months post-conization.
Per the national guidelines, the initial six-month follow-up after conization typically involves a co-test that includes both liquid-based cytology (LBC), usually using the ThinPrep system, and an HPV test conducted with the Roche Cobas 4800 system. If the co-test results are negative, the patient is advised to return to the regular screening schedule, which occurs every three years.
If the HPV test is positive but the cytology is normal, the follow-up protocol involves repeating both the cytology and the HPV test after an additional six months. Conversely, if the cytology is abnormal and the HPV test is positive, the patient is then referred for colposcopy and biopsy as needed.
In our study, we focused on HPV status as the primary endpoint rather than biopsy results. This decision was influenced by the fact that most women did not undergo a biopsy after treatment, in line with the Norwegian national guidelines for post-treatment follow-up, which prioritize non-invasive monitoring unless cytological abnormalities persist or reappear.
In the methods section we have now added: "While Norwegian guidelines recommend co-testing with both cytology and HPV test at the first follow-up after treatment, this study focused solely on the post-conization HPV results as the study endpoint."
We hope this clarifies the follow-up procedures used in our study and the rationale behind our focus on HPV status.